# Association of metformin use with risk and survival outcome of esophageal cancer in patients with diabetes: A systematic review and meta-analysis

Hui Xie[1,2]☯, Muhan Li[3]☯, Zhaoqi Chen●[4]*, Yuling Zheng[1,2]*

**1** The First Affiliated Hospital of Henan University of Chinese Medicine, Zhengzhou, China, **2** The First Clinical Medical College of Henan University of Chinese Medicine, Zhengzhou, China, **3** The First Clinical Medical College, Nanjing University of Chinese Medicine, Nanjing, China, **4** Henan Provincial People's Hospital, Zhengzhou, China

☯ These authors contributed equally to this work.

* 370411478@qq.com

**Data Availability Statement:** All relevant data are within the manuscript.

**Funding:** This work was funded by 2022 National Famous Traditional Chinese Medicine Experts'

## Abstract

The purpose of this study was to conduct a systematic review and meta-analysis to investigate the association between the utilization of metformin and the occurrence and survival rate of esophageal cancer (EC) in individuals with diabetes. Methods: A systematic review and a meta-analysis were performed. Related literature was searched from databases, including PubMed, Embase, Web of Science, and Cochrane Library, covering the period from the inception of these databases until July 2023. Results: A total of 16 studies were eligible, including twelve reporting incidences of EC and four reporting OS of EC patients. The combined findings revealed a significant association between the use of metformin and a lower risk of EC (OR, 0.87, $P = 0.04$). Furthermore, metformin could significantly prolong the OS time (HR, 0.87, $P = 0.002$). In analyses stratified by treatment modalities, metformin combined with surgery and neoadjuvant chemoradiotherapy presented the strongest protective effect on EC patients with diabetes (HR, 0.38, $P = 0.003$). Conclusion: Our meta-analysis indicated that the use of metformin might reduce the EC incidence and improve the OS in EC patients with diabetes.

## Introduction

Esophageal cancer (EC) represents the seventh most common malignancy worldwide with a 5-year survival rate less than 20% [1,2]. It is ranked as the sixth leading cause of cancer-related death in the world [3]. Over the past few decades, there has been a significant rise in the occurrence of esophageal adenocarcinoma in Western nations [4–6]. Concurrently, the prevalence of diabetes has been steadily increasing [7]. Growing evidence indicates a close relationship between the risk of cancer and type 2 diabetes [8–10].

Inheritance Studio (G.Z.Y.J.H. [2022] No. 245), 2022 National Famous old Traditional Chinese Medicine Experts' Inheritance Studio (G.Z.Y.J.H. [2022] No. 75), and Collaborative Innovation Center of Prevention and Treatment of Major Diseases by Chinese and Western Medicine, Henan Province [2023] 413. Funders did not contribute to the design, data collection, analysis, preparation of the manuscript, and final publication decision.

**Competing interests:** The authors have declared that no competing interests exist

Metformin, a first-line agent utilized for diabetes, has demonstrated potential in reducing the risk to diverse malignancies [11–15]. A meta-analysis published in June 2020 [16] has demonstrated that metformin does not reduce EC risk in patients with diabetes (HR 0.88, $P>0.05$). In this study, however, the relationship between metformin and the prognosis of EC was not reported, and subgroup analysis according to treatment methods and EC pathological types was not carried out either. We therefore conducted a meta-analysis to further illustrate the association of metformin with the risk and overall survivals in EC patients.

## Materials

The present study adhered to the Preferred Reporting Items for Systematic Reviews and Meta-Analysis (PRISMA) guidelines [17] (S1 Checklist). Prior to commencing the study, it was registered on PROSPERO (registration no. CRD42023411063).

### Search strategy

A comprehensive systematic review was undertaken to identify relevant studies pertaining to EC, encompassing the period from their inception up until July 2023. The adjusted odds ratio (OR) for the incidence or adjusted hazard ratios (HRs) for the overall survival (OS) of EC patients with diabetes who received metformin were compared to those of who did not. The databases PubMed, Embase, Cochrane Library, and Web of Science were meticulously searched for original articles. All the included literatures were published in English. The search utilized specific keywords and medical terms, namely "Metformin," "Glucophage," "Dimethyl biguanide," "Esophageal Neoplasms," "Esophagus Cancer," "Cancer, Esophageal," "Neoplasms, Esophagus," "Diabetes Mellitus," and "Prediabetic State."

### Inclusion and exclusion criteria

1. Metformin was administered to prevent or treat EC in diabetes patients.

2. Studies provided data on the incidence of EC or overall survival (OS) or all-cause mortality.

3. Studies directly reported effect size such as odds ratio (OR), hazard ratio (HR) with 95% confidence intervals.

4. The studies were observational design.

The publications were excluded if they met any of the following criteria:

1. Therapy with drugs other than metformin.

2. Absent focus on the association between metformin and EC.

3. Not human-based studies.

4. Case reports, letters, commentaries, reviews, and unpublished data were excluded.

### Data collection and quality assessment

The eligibility of studies was evaluated, and pertinent data was extracted by two reviewers (Xie and Li) independently. Any discrepancies were resolved through consensus. Standardized tables were generated to extract key information from each study, involving age, author name, publication year, country, study design, sample size, follow-up period, outcome, and effect size. If the same study population appeared in multiple studies, the most recent or

comprehensive one was included. The prioritization for analysis involved adjusted hazard ratios (HRs) or odds ratios (ORs) accompanied by 95% confidence intervals (CIs). The raw extracted data from the included studies can be found in S1 Table. The reviewers autonomously evaluated the quality and potential bias of each study using the Newcastle-Ottawa Scale [18]. This is detailed in S2 Table.

## Statistical analysis

The RevMan software v5.3 was employed to conduct this meta-analysis. ORs, HRs and their corresponding 95% confidence intervals (CIs) were calculated to assess the risk and prognosis of EC. A OR or HR value below 1 considered to associate with a decreased risk of disease and a better survival in patients using metformin. Heterogeneity was assessed using $I^2$ and Cochran Q estimates, with $I^2$ values exceeding 50% or p values below 0.1 indicative of significant heterogeneity. The DerSimonian-Laird random-effects model was employed for calculating the pooled OR or HR directly, while an inverse variance fixed-effects model was utilized when applicable.

Forest plots were generated using Review Manager Software v5.3. The sensitivity analysis was performed by systematically excluding one study at a time and re-analyzing the remaining studies to ascertain the influence of each study on the overall findings. Publication bias was assessed using the funnel plot, as well as the Begg and Egger tests, using the StataSE 16 software. Furthermore, the subgroup analysis was conducted to examine potential biases associated with region, pathological type, sample size, body mass index (BMI), sex, age, treatment, and comparators. Statistical significance for all tests was determined at a significance level of $p < 0.05$.

## GRADE

The GRADE was performed to evaluate the overall of evidence across studies [19]. The level of evidence quality was divided into four categories: high (4), moderate (3), low (2), and very low (1). Evidence quality can indicate the confidence of effect estimates. The GRADE system has outlined five factors contributing to a decrease in evidence quality, including study design, risk of bias, imprecision, inconsistency, indirectness, and magnitude of effect [19]. Conversely, an increase in evidence quality is associated with a substantial effect size, a dose-response relationship, and a high plausibility of the findings.

# Results

## Studies included

A flow diagram detailing the process of identification, inclusion, and exclusion of publications is shown in Fig 1. From all the publications searched (11 from PubMed, 146 from Embase, 1 from the Cochrane Library, 56 from Web of Science), 50 articles were removed as duplicates, leaving 164 articles to be screened based on abstracts and titles. Of these, 146 publications, involving animal or cell experiments, reviews, meta-analyses, conference reports, and irrelevant studies, were excluded. Then, Then, two articles without the outcome indication were eliminated (S3 Table). Finally, 16 eligible studies were pooled into the meta-analysis (S4 Table).

## Study characteristics

Of the 16 included studies, 14 were cohort studies, and 2 was a case-control study (Table 1). All analyses were published between 2011 and 2023. One study was conducted in America [20], 9 in Europe (4 in the Netherlands, 2 in the UK, 2 in Sweden, and 1 in Italia) [21–29], and

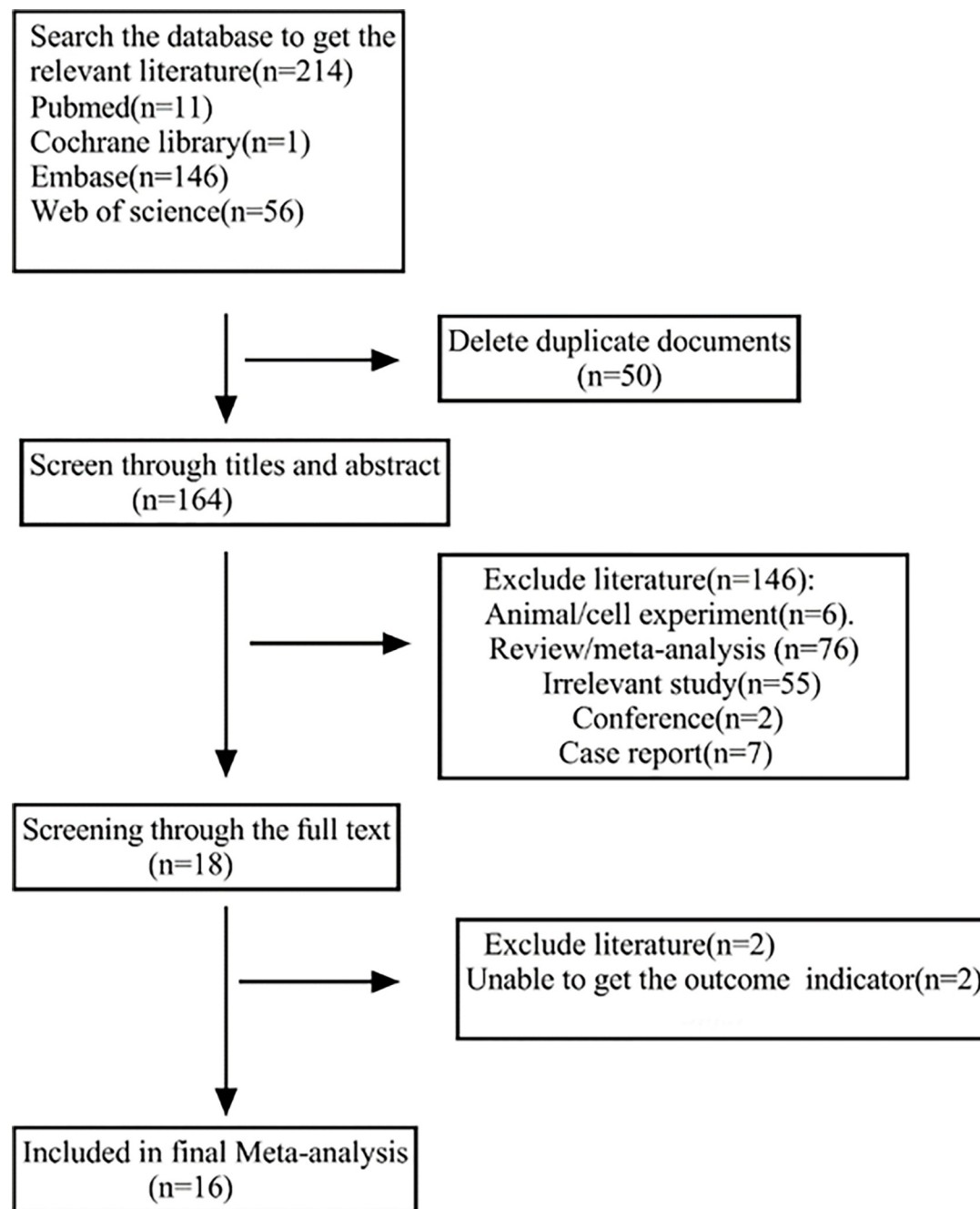

**Fig 1. Study flow chart for the process of selecting the final 16 publications.**

6 in Asia (5 in China, 1 in South Korea) [30–35]. EC incidence was analyzed in 12 studies, overall survival in 4 studies. The sizes of study populations ranged from 43 to 4,527,633. Incidence rate Metformin was used for at least 9.6 months. Incidence of EC ranged from 3.5 to 376 per 100,000 person-years in the metformin group and from 5.3 to 457 per 100,000 person-years in the no-metformin group. Mortality from EC ranged from 21 to 79.9% in the metformin group and from 50.3 to 90.7% in the no-metformin group. Among the studies on survival of patients with esophageal cancer, three studies mainly included patients with esophageal

**Table 1. General characteristics of included studies.**

| First author, publication year, country | Study Design | Duration of metformin use (months) | Total Sample Size | Number of patients (Met/ NM) | Incidence rate (per 100,000 person-years) (Met/ NM) | Mortality rate (Met/ NM) | NOS score | Stage | Histopathology | Outcomes | Reported OR/HR (95% CI) |
|---|---|---|---|---|---|---|---|---|---|---|---|
| Qiao-Li Wang [21], 2019, Sweden | CS | - | 4,527,633 | 5411,603/ 4116030 | 3.5 /5.3 | - | 8 | - | ESCC | IEC | 0.68[0.54, 0.85] |
| Chin-Hsiao Tseng [34], 2016, China | CS | < 21.47 | 304,229 | 16216/ 16216 | 25.03/50.87 | - | 8 | - | - | IEC | 0.56[0.33, 0.94] |
| Joseph JY Sung [30], 2020, China | CS | 56.4 | 289,297 | 11,365/ 277,932 | - | - | 7 | - | - | IEC | 0.27[0.12, 0.59] |
| Konstantinos K. Tsilidis [23], 2014, UK | CS | 12 | 95,820 | 51484/ 18264 | - | - | 8 | - | - | IEC | 1.05[0.71, 1.56] |
| Meei-Shyuan Lee [31], 2011, China | CS | 24 | 480,984 | 11212/ 4193 | 47.8/40.3 | - | 7 | - | - | IEC | 0.44[0.07, 2.61] |
| Harvey J. Murff [20], 2018, USA | CS | 24 | 84,434 | 42217/ 42217 | 50/50 | - | 6 | - | - | IEC | 0.99[0.63, 1.55] |
| Roy G. de Jong [25], 2017, Netherlands | CS | - | 57,114 | 37215/ 19899 | 376/457 | - | 8 | - | - | IEC | 0.90[0.48, 1.67] |
| Rikje Ruiter [26], 2012, Netherlands | CS | 12 | 85,289 | 52,698/ 32,591 | 30/30 | - | 7 | - | - | IEC | 0.90 [0.82, 0.97] |
| Tak Kyu Oh [33], 2019, South Korea | CS | 9.6 | 66,627 | 29974/ 36653 | - | - | 7 | - | - | IEC | 0.38 [0.13, 1.13] |
| Claudia Becker [22], 2013, UK | CCS | >60 | 4,070 | 2040/2030 | - | - | 6 | - | - | IEC | 1.11[0.79, 1.54] |
| Francesca Valent [29], 2015, Italian | CS | - | 109255 | 63119/ 75402 | - | - | 7 | - | - | IEC | 0.98 [0.98,1.00] |
| Kao-Chi Cheng [35], 2012, China | CCS | - | 279 | 231/48 | - | - | 6 | - | - | IEC | 2.84 [0.99,8.18] |
| L. Van De Voorde [24], 2015, Netherlands | CS | - | 196 | 19/177 | - | 21%/50.3% | 6 | T3 (68%) N0-N1 (80%) | Adenocarcinoma (78.1%) | OS | 0.35 [0.13, 0.97] |
| Huang-He He [32], 2020, China | CS | - | 619 | 485/134 | - | 69.07%/ 77.61% | 8 | II-III (77.7%) | ESCC (100%) | OS | 0.89 [0.80, 0.99] |
| L. E. A. M. M. Spierings [27], 2015, Netherlands | CS | - | 43 | 32/11 | - | - | 6 | T3 (77.7%) N1-N2 (71.1%) | Adenocarcinoma (68.3%) | OS | 0.49 [0.08, 2.92] |
| Qiaoli Wang [28], 2023, Sweden | CS | 12 | 852 | 473/379 | - | 79.9%/ 90.7% | 8 | III-IV (59.3%) | Adenocarcinoma (57.6%) ESCC (37.0%) | OS | 0.86 [0.75,1.00] |

ESCC, Esophageal squamous cell carcinoma; Met, Metformin; NM, Non-metformin; CS, Cohort study; CCS, Case control study; IEC, Incidence of esophageal cancer; OS, Overall survival; NA, Not available; EC, Esophageal cancer.

adenocarcinoma, one study focused on esophageal squamous-cell carcinoma (ESCC). The NOS scores of all included studies ranged between 6 and 8, with an average of 7.06, which indicated that methodological quality was generally acceptable.

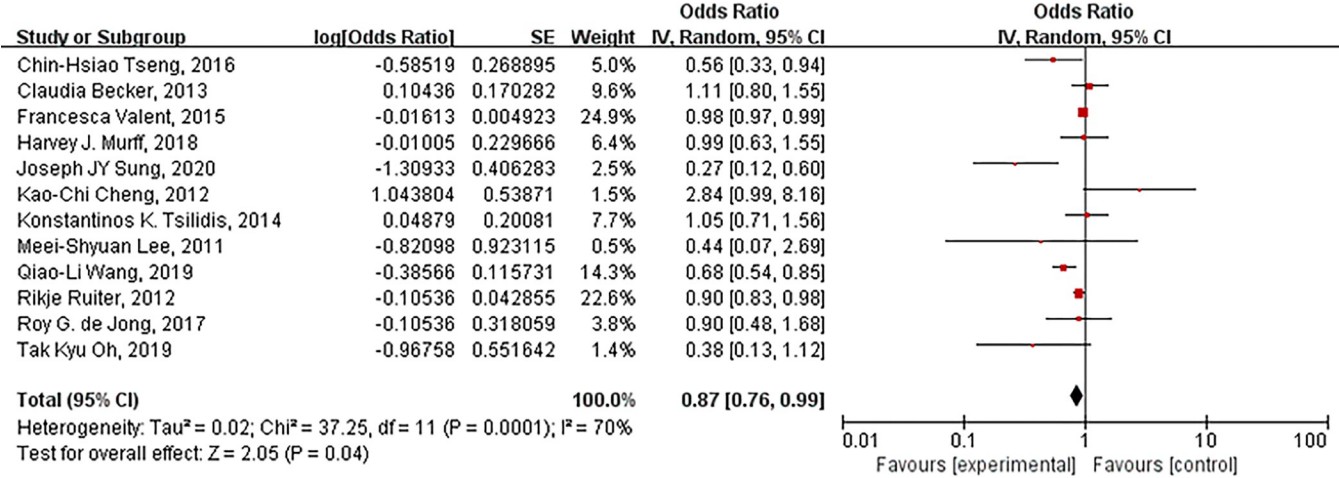

**Fig 2. Forest plot of the association between metformin use and risk of EC among patients with diabetes.**

## Association of metformin with the EC incidence in patients with diabetes

A total of 12 studies were selected into the meta-analysis (Fig 2). The random effects model showed that compared with the control groups, metformin might decrease EC incidence (OR = 0.87, $P$ = 0.04). The heterogeneity test showed moderate heterogeneity among these studies (Heterogeneity: $I^2$ = 70%, $P$ = 0.0001). The sensitivity analysis showed that the combined OR was relatively stable. The Begg funnel plot showed the basically symmetric distribution (Fig 3). Egger's ($P$ = 0.373) and Begg's bias test ($P$ = 0.251) showed no potential publication bias.

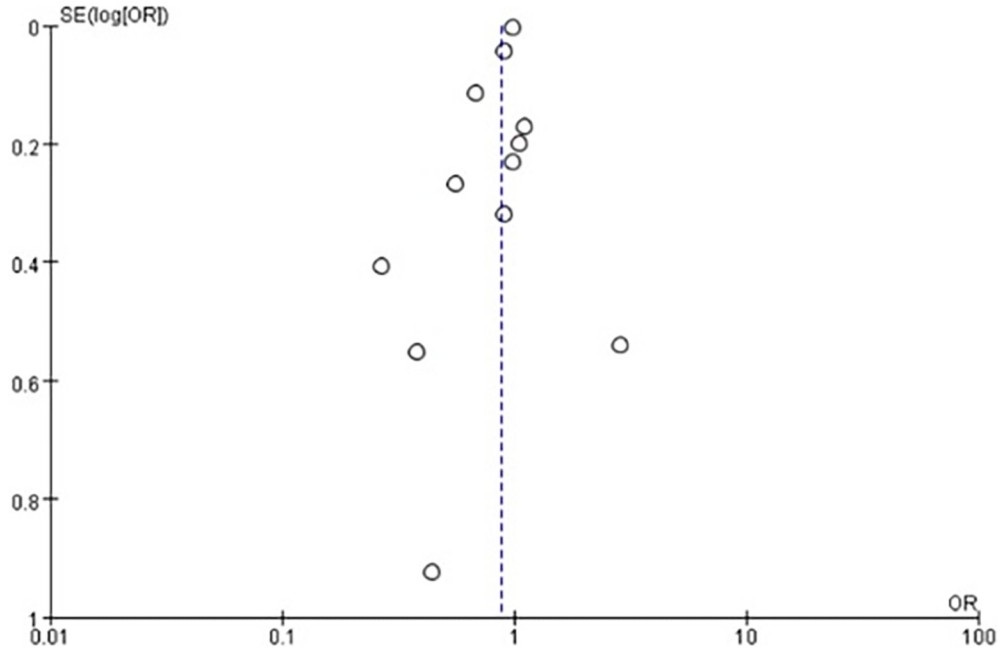

**Fig 3. Begg's funnel plot for EC risk in patients with diabetes.**

**Table 2. Summary of secondary analyses for metformin and incidence of EC.**

| Subgroup analyses | | Number | Heterogeneity | | Model | Meta-analysis | |
|---|---|---|---|---|---|---|---|
| | | | $p$ | $I^2$ | | HR (95% CI) | $P$ |
| Region | Asia | 4* | 0.52 | 0% | Fixed | 0.44 [0.29, 0.65] | <0.0001 |
| | Europe and America | 6# | 0.42 | 0% | Fixed | 0.98 [0.97, 0.99] | 0.0004 |
| Comparators | Sulfonylureas | 3 | 0. 7 | 0% | Fixed | 0.91 [0.84, 0.99] | 0.02 |
| | Non-metformin | 9 | <0.0001 | 76% | Random | 0.78 [0.59, 1.02] | 0.07 |
| Time-related biases | Yes | 5 | 0.01 | 69% | Random | 0.94 [0.84, 1.06] | 0.33 |
| | No | 6& | 0.11 | 44% | Fixed | 0.81 [0.68, 0.95] | 001 |
| BMI | >30 | 2 | 0.85 | 0% | Fixed | 1.02 [0.76, 1.38] | 0.88 |
| Use of aspirin | Yes | 4 | 0.14 | 45% | Fixed | 0.76 [0.63, 0.91] | 0.004 |
| Use of statins | Yes | 3 | 0.15 | 47% | Fixed | 0.77 [0.64, 0.93] | 0.006 |
| Use of Insulin | Yes | 2 | 0.4 | 0% | Fixed | 0.49 [0.21, 1.11] | 0.09 |
| Mean age | <60 | 3 | 0.72 | 0% | Fixed | 0.66 [0.53, 0.81] | <0.0001 |
| | 60≥ | 7& | 0.18 | 33% | Fixed | 0.92 [0.85, 0.99] | 0.03 |
| BMI | No adjusted | 9 | <0.0001 | 78% | Random | 0.81 [0.69, 0.95] | 0.01 |
| | adjusted | 3 | 0.92 | 0% | Fixed | 1.06 [0.85, 1.32] | 0.6 |
| Smoking | No adjusted | 8 | 0.0004 | 74% | Random | 0.86 [0.73, 1.01] | 0.07 |
| | Adjusted | 3# | 0.92 | 0% | Fixed | 1.06 [0.85, 1.32] | 0.6 |
| Sample size | ≥300000 | 3 | 0.72 | 0% | Fixed | 0.66 [0.53, 0.81] | <0.0001 |
| | <300000 | 8& | 0.11 | 41% | Fixed | 0.98 [0.97, 0.99] | 0.0004 |

Abbreviations: HR hazard ratio; CI confidence interval; BMI body mass index.

*Removed Kao-Chi Cheng's study; #Removed Qiao-Li Wang's study; &Removed Joseph JY Sung's study.

Table 2 summarizes secondary and exploratory analyses. The stratification analysis by eth-nicity found that metformin was associated with a reduction in the EC incidence in Asian and Europe and America, more significantly in Asian populations (OR,0.44, $P$<0.0001 vs OR,0.98, $P$ = 0.0004). In the groups with BMI>30, metformin could not reduce the risk of EC (OR, 1.02, $P$ = 0.88). In the studies, metformin combined with aspirin or statin, significantly reduced EC incidence (OR, 0.76, $P$ = 0.004; OR, 0.77, $P$ = 0.006). However, its combination with insulin did not inhibit the development of EC (OR, 0.49, $P$ = 0.09). Compared to those adjusted for BMI, those studies not adjusted for BMI showed that metformin had a stronger protective effect on EC patients with diabetes (OR, 1.06, $P$ = 0.6; OR, 0.81, $P$ = 0.01). The details are illustrated in Table 2.

## Association of metformin with the OS in EC patients with diabetes

There was low heterogeneity among the included 4 studies ($P$ = 0.3 and $I^2$ = 18%). To assess whether our findings were stable, sensitivity analysis was conducted by sequentially omitting a single study at a time. The sensitivity analysis showed a stable outcome. The results showed that metformin could increase the OS of EC patients with diabetes (HR 0.87, $P$ = 0.002), as shown in Fig 4. We performed Egger's and Begg's funnel plots to check publication bias. Egger's ($P$ = 0.106) and Begg's ($P$ = 0.308) bias tests showed no potential publication bias.

A summary of secondary analyses is described in Table 3. In either European and American or Asian populations, metformin significantly prolonged the survival time of EC patients (HR 0.84, $P$ = 0.02; HR 0.89, $P$ = 0.03). In the subgroup analysis according to histopathological stage, metformin retained its effect only in squamous cell carcinoma patients (HR 0.89, $P$ = 0.02). The protective effect of metformin was closely related to gender and age. Metformin

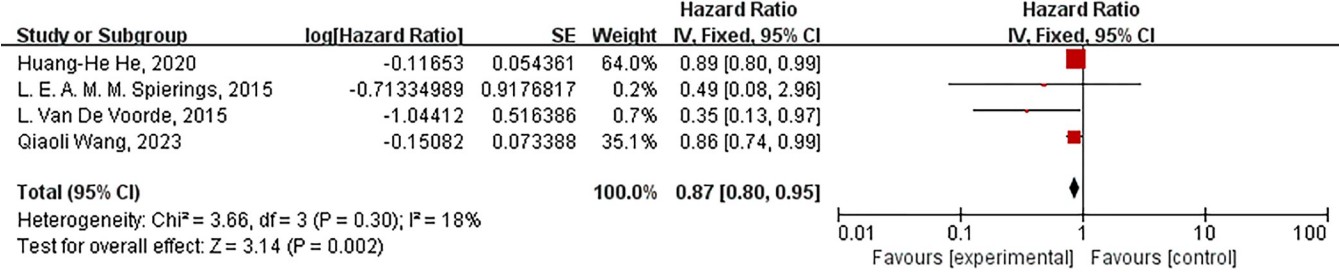

**Fig 4. Forest plot of metformin and OS.**

had a more significant protective effect in female populations or those under the age of 70(HR 0.88, $P$ = 0.01; HR 0.88, $P$ = 0.02). In the subgroup analysis according to treatment modality, metformin with surgery, or with surgery and neoadjuvant radiotherapy and chemotherapy, or with statins exerted more significant protective effects in EC patients than in those who did not use metformin (HR 0.89, $P$ = 0.02; HR 0.38, $P$ = 0.03; HR 0.86, $P$ = 0.03). Moreover, metformin combined with surgery and neoadjuvant chemoradiotherapy presented the strongest protective effect on EC patients with diabetes. The protective effects of metformin remained significant in the subgroups stratified by BMI, smoking, and sample size. See Table 3 for more details.

As determined by the GRADE system, the quality of evidence concerning the incidence was very low and OS was low (Table 4). First, they were observational studies. Second, too few participants were included, especially for OS. Third, confounders like tumor stage, mean

**Table 3. Summary of secondary analyses for metformin and overall survival.**

| Subgroup analyses | | Number | Heterogeneity | | Model | Meta-analysis | |
|---|---|---|---|---|---|---|---|
| | | | $p$ | $I^2$ | | HR (95% CI) | $P$ |
| Region | Europe and America | 3 | 0.19 | 39% | Fixed | 0.84 [0.73, 0.97] | 0.02 |
| | Asia | 1 | - | - | - | 0.89 [0.80, 0.99] | 0.03 |
| Histology | Adenocarcinoma | 3 | 0.18 | 42% | Fixed | 0.85 [0.72, 1.01] | 0.07 |
| | Squamous cell carcinoma | 2 | 0.77 | 0% | Fixed | 0.89 [0.80, 0.98] | 0.02 |
| Age | >70 | 1 | - | - | - | 0.84 [0.69, 1.03] | 0.09 |
| | ≤70 | 3 | 0.17 | 44% | Fixed | 0.88 [0.79, 0.98] | 0.02 |
| Sex | Predominantly male | 3 | 0.19 | 40% | Fixed | 0.85 [0.72, 1.00] | 0.05 |
| | Predominantly Female | 2 | 0.59 | 0% | Fixed | 0.88 [0.80, 0.98] | 0.01 |
| BMI | <30 | 3 | 0.11 | 44% | Fixed | 0.88 [0.79, 0.98] | 0.02 |
| Without Surgery | Yes | 1 | - | - | - | 0.87 [0.75, 1.01] | 0.08 |
| Surgery | Yes | 4 | 0.25 | 26% | Fixed | 0.89 [0.80, 0.98] | 0.02 |
| Surgery + NC | Yes | 2 | 0.75 | 0% | Fixed | 0.38 [0.16, 0.92] | 0.03 |
| Use of statins | Yes | 2 | 0.54 | 0% | Fixed | 0.86 [0.74, 0.99] | 0.03 |
| BMI | No adjusted | 3 | 0.19 | 39% | Fixed | 0.84 [0.73, 0.97] | 0.02 |
| | adjusted | 1 | - | - | - | 0.89 [0.80, 0.99] | 0.03 |
| Smoking | No adjusted | 3 | 0.19 | 39% | Fixed | 0.84 [0.73, 0.97] | 0.02 |
| | adjusted | 1 | - | - | - | 0.89 [0.80, 0.99] | 0.03 |
| Sample size | <500 | 2 | 0.75 | 0% | Fixed | 0.38 [0.16, 0.92] | 0.003 |
| | ≥500 | 2 | 0.71 | 0% | Fixed | 0.88 [0.81, 0.96] | 0.003 |

Abbreviations: HR, hazard ratio; CI, confidence interval; BMI, body mass index; NC, neoadjuvant chemoradiotherapy.

**Table 4. Association of metformin with IEC, OS, and GRADE evidence profile.**

| Outcome indicators | No. of studies | Certainty assessment | | | | | | | | Effect | |
|---|---|---|---|---|---|---|---|---|---|---|---|
| | | Study design | Risk of bias | Inconsistency | Indirectness | Imprecision | Publication bias | Effect size | dose response | HR (95%CL) | Certainty |
| IEC | 12 | CS | Serious | Serious | NS | Serious | No | Small | No | 0.85 (0.72,0.99) | Very low |
| OS | 5 | CS | Serious | NS | NS | NS | No | Small | No | 0.87 (0.80,0.95) | Low |

CS, Cohort study; NS, Not Serious; IEC, Incidence of esophageal cancer; OS, Overall survival.

metformin duration, complications, and antitumor drugs, were not controlled. Fourth, no dose-response analysis was conducted.

## Discussion

In the current study, we conducted a systematic review and two separate meta-analyses on the association of metformin with EC incidence and OS in patients with diabetes. We found a significant relationship between metformin use and a low incidence of EC in twelve pooled studies (OR = 0.85, $P$ = 0.04). In our meta-analysis regarding metformin exposure and OS, metformin prolonged the OS time by 13% in five pooled studies (HR 0.87, $P$ = 0.002). Rated by the GRADE, the overall qualities of evidence in both meta-analyses were "very low"; therefore, the estimated effect of metformin should be considered as uncertain and needs further investigations.

For EC risk, our finding is incongruous with two previous meta-analyses conducted by H.-D. Wu et al. [16] but aligned with the results from the research by Kui Zhang et al. [36] and Monica Franciosi et al. [37]. We observed that only four studies published prior to 2020 were incorporated into Wu's meta-analysis, five into Zhang's, and two into Franciosi's. The data extracted by WU was not propensity score (PS) adjusted, while our data was after the PS matched. As is well known, PS matching can reduce the impact of confounding factors. When we conducted a meta-analysis using the data from Wu's study, we found that heterogeneity was obvious declined after excluding Chin Hsiao Tseng's study ($I^2$ from 52% to 2%) [34]. However, Wu did not exclude this literature and analyze the reason. Wu et al. only deleted Claudia Becker's study [22] based on as a case-control and calculated the risk of EC just based on HR values, which is not reasonable. Zhang et al. did not conduct an independent meta-analysis specific for EC. Zhang's meta-analysis had moderate heterogeneity ($I^2$ = 57.9%) and did not analyze the sources of heterogeneity. In the present study, twelve original studies were included, and subgroup analyses were conducted, which enhances the reliability of our research results.

Our subgroup analysis demonstrated a slight disparity in the protective effects of metformin between Asians, Europeans, and Americans (Asians OR, 0.44, $P$<0.0001; Europeans and Americans OR, 0.98, $P$ = 0.0004). The reason may be attributed to the following factors. First,

EC has a higher incidence in Asian than in most Western countries. Especially incidence of squamous cell carcinoma has been traditionally much higher in regions of China [38,39]. Second, there exist variations in risk factors across different regions. In Europe and America, alcohol consumption and smoking account for the EC occurring in approximately 90% of the cases [40,41], whereas undernutrition, low intake of fruits and vegetables, high intake of salted and preserved foods, and hot beverages are identified as high-risk factors of EC in Asians [42–44]. Four studies targeting Asian populations did not exclude these high-risk factors

[30,31,34,35]. Of the six studies targeting European and American populations, smoking was ruled out in four studies [20–23] and alcohol consumption in two studies [22,23]. Third, gastroesophageal reflux disease, obesity, alcohol, and metabolic syndrome are common causes of EAC in Europe and America [45,46]. These populations are also the primary consumers of metformin [47], which possibly leading to overestimation of EAC risk. Consequently, we speculate that geographical prevalence and risk factors may elucidate the inconsistent effects of metformin on the risk of EC.

Numerous studies have investigated the correlation between BMI and EC. Engeland et al. [48] have found that a high increase the risk of EAC, which is in agreement with our present finding that the participants with BMI >30 had a higher risk of EC (OR 1.02, $P = 0.88$). Patients with diabetes are often accompanied by dyslipidemia, atherosclerosis and cardiovascular diseases, and more likely to seek statin and aspirin treatment [49,50]. In our meta-analysis, metformin combined with statins or aspirin could reduce the risk of EC in patients with diabetes (OR 0.77, $P = 0.006$; OR 0.76, $P = 0.004$). Vijayvel Jayaprakash et al. [51] have also found that the effect of aspirin is more pronounced in reducing the risk of adenocarcinoma than that of squamous cell carcinoma. Regular use of aspirin is associated with a 50% reduction in EAC risk and 37% reduction in the risk of esophageal squamous cell carcinoma [51]. The induction of cancer cell apoptosis and inhibition of inflammatory response by aspirin may serve as the underlying mechanism by which aspirin disrupts esophageal carcinogenesis [52,53]. In our study, metformin combined with insulin could not reduce the risk of EC (OR 0.49, $P = 0.09$). Because insulin is more likely to be used by patients with severe and long-term symptoms, this population may face a higher risk of cancer [54].

We further found that use of metformin was associated with an almost 13% improvement in the OS among EC patients with diabetes (HR 0.87, $P = 0.002$). The subgroup analysis of treatment modality indicated that metformin combined with surgery and neoadjuvant chemoradiotherapy could improve patients' OS, compared to metformin alone without surgery (HR,0.38, $P = 0.03$ vs HR,0.87, $P = 0.08$). In recent years, the need for neoadjuvant chemoradiotherapy has increasingly been recognized in esophageal cancer. Several studies have reported that its use has improved local control, progression free survival, and OS in comparison with surgery alone [55,56]. There are multiple advantages to this therapy. First, neoadjuvant chemoradiotherapy can downstage tumors, improving their respectability and decreasing the rate of positive surgical margins. Second, the necrosis and fibrosis of tumor tissue after neoadjuvant chemoradiotherapy greatly reduces active tumor cells and the probability of tumor cells shedding, spreading, and planting during surgery, thereby reducing local recurrence rates [55,57,58].

Our observations further support that metformin may possess antitumor properties in individuals afflicted with EC. The underlying mechanism may involve the ability of metformin in increasing the sensitivity of liver, muscle, fat, and other tissues to insulin, thereby lowering blood sugar levels and reducing insulin resistance [59,60]. Recent research suggests that metformin effectively suppresses the proliferation of esophageal squamous cell carcinoma-TE1 and Ecal09 cells. Metformin enhances the radiosensitivity of TE-1 and ECal09 esophageal squamous carcinoma cell lines [61]. Metformin has been found to upregulate AMPK, thereby inhibiting various metabolic processes that heavily rely on adenosine triphosphate (ATP) in cells, including protein and fatty acid synthesis, as well as promoting cellular catabolic processes, such as fatty acids β catabolism and, glycolysis Metformin-activated AMPK also inhibits multiple mTOR signaling pathways, thereby impeding abnormal signal transduction from growth factors, subsequently hindering mRNA translation, and ultimately preventing the development and occurrence of tumors [62–64]. In the subgroup analysis of pathological type, although the protective effect of metformin may have been more pronounced for squamous

cell carcinoma than for adenocarcinoma (HR 0.89, $P$ = 0.02; HR 0.85, $P$ = 0.07). However, the number of cases in each category of histological subtypes was too small to arrive at any reasonable conclusion regarding the difference between the histological types.

There are several limitations to the current analyses. Initially, all the studies were observational, potentially introducing recall and selection biases. Furthermore, this study did not analyze the confounding variables that may link with the survival rates of EC patients, such the treatment frequency, dosage, and duration of metformin usage, and the stage of cancer. The weight of individual studies in some subgroup analyses was too large; therefore, the results should also be interpreted with caution. In addition, the quality of evidence was considered low by the GRADE system, so some conclusions are controversial.

Nevertheless, our research also exhibits its strengths. First, it included a greater number of primary studies and conducted sensitivity and publication bias analyses. Additionally, the subgroups were well designed according to multiple factors.

## Conclusion

Metformin may have an inhibitory impact on EC occurrence and a positive impact on the OS in EC patients with diabetes. Metformin combined with surgery and neoadjuvant radiotherapy may improve OS more effectively, compared to metformin alone or just combined with surgery. Metformin may be recommended as an effective treatment to EC patients with diabetes. Nevertheless, it should interpret our results with caution, due to potential methodological biases and the low-quality evidence. Furthermore, a prospective randomized controlled study with a larger sample is needed to warrant the efficacy of metformin combined surgery and neoadjuvant chemoradiotherapy in the treatment of EC.

## Supporting information

**S1 Checklist. PRISMA 2020 checklist.**
(DOCX)

**S1 Table. Raw data for meta-analysis.**
(DOCX)

**S2 Table. Newcastle-Ottawa quality assessment scale.**
(DOC)

**S3 Table. Articles with reasons for exclusion.**
(DOCX)

**S4 Table. Studies included in the meta-analysis.**
(DOCX)

## Acknowledgments

We thank Pomegranate Translation Studio (Nanjing, China), for its professional editing on this manuscript.

## Author Contributions

**Conceptualization:** Hui Xie, Zhaoqi Chen.

**Data curation:** Hui Xie.

**Formal analysis:** Muhan Li.

**Funding acquisition:** Yuling Zheng.

**Investigation:** Hui Xie.

**Methodology:** Muhan Li.

**Project administration:** Muhan Li.

**Software:** Hui Xie, Muhan Li, Yuling Zheng.

**Supervision:** Muhan Li, Zhaoqi Chen.

**Visualization:** Hui Xie, Yuling Zheng.

**Writing – original draft:** Hui Xie.

**Writing – review & editing:** Zhaoqi Chen, Yuling Zheng.

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
