## [Decision Letter · Decision Letter 0]

22 Jul 2024

PONE-D-24-24173Association of metformin use with risk and survival outcome of esophageal cancer in patients with diabetes: a systematic review and meta-analysis.PLOS ONE

Dear Dr. Chen,

Thank you for submitting your manuscript to PLOS ONE. After careful consideration, we feel that it has merit but does not fully meet PLOS ONE’s publication criteria as it currently stands. Therefore, we invite you to submit a revised version of the manuscript that addresses the points raised during the review process.

We look forward to receiving your revised manuscript.

Kind regards,

Jaspinder Kaur, MD

Academic Editor

PLOS ONE

Journal Requirements:

2. PLOS requires an ORCID iD for the corresponding author in Editorial Manager on papers submitted after December 6th, 2016. Please ensure that you have an ORCID iD and that it is validated in Editorial Manager. To do this, go to ‘Update my Information’ (in the upper left-hand corner of the main menu), and click on the Fetch/Validate link next to the ORCID field. This will take you to the ORCID site and allow you to create a new iD or authenticate a pre-existing iD in Editorial Manager. Please see the following video for instructions on linking an ORCID iD to your Editorial Manager account: https://www.youtube.com/watch?v=_xcclfuvtxQ"

Reviewers' comments:

Reviewer's Responses to Questions

**Comments to the Author**

1. Is the manuscript technically sound, and do the data support the conclusions?

Reviewer #1: No

Reviewer #2: Yes

2. Has the statistical analysis been performed appropriately and rigorously? 

Reviewer #1: Yes

Reviewer #2: Yes

3. Have the authors made all data underlying the findings in their manuscript fully available?

Reviewer #1: Yes

Reviewer #2: Yes

4. Is the manuscript presented in an intelligible fashion and written in standard English?

Reviewer #1: Yes

Reviewer #2: Yes

5. Review Comments to the Author

Reviewer #1: 1.Please indicate reference number for each study in table 1. Please provide the breakdown, indicating the number of patients/ participants on Metformin vs no Metformin. Currently on table 1, there is only one group as NM (Non-Metformin).

2.What was the absolute number of new cases of esophageal cancer in the group exposed to Metformin and non-Metformin, in each study where incidence was the outcome of interest and combined.

3.What was the stage of the disease and histopathology in each study where a therapeutic intervention was done.

4.Authors have provided significant statistical differences without any absolute numbers of an outcome/ event related to esophageal cancer or clinical characteristics of the patients exposed to Metformin or no Metformin, sulphonylureas or no sulphonylureas.

5.Given the huge sample size in the analyzed studies and without detailed clinical information of the participants, incidence/ extent of esophageal cancer it is challenging to make a sense of statistical values provided in the manuscript.

Reviewer #2: First, I would like to praise your effort in this metanalysis and data collection and there is one main point for me need clarification

Heterogenicity test index in association of metformin with the EC incidence in patients with diabetes is near the high index and the subgroup analysis did not eliminate its effect and did not answer questions like how long did patients suffer from diabetes before EC and how long did they use metformin before diagnosis by EC

6. PLOS authors have the option to publish the peer review history of their article (what does this mean?). If published, this will include your full peer review and any attached files.

Reviewer #1: No

Reviewer #2: **Yes: **Doaa Aly

---

## [Author Response · Author response to Decision Letter 0]

2 Sep 2024

Reviewer #2

 First, I would like to praise your effort in this metanalysis and data collection and there is one main point for me need clarification 

Heterogenicity test index in association of metformin with the EC incidence in patients with diabetes is near the high index and the subgroup analysis did not eliminate its effect and did not answer questions like how long did patients suffer from diabetes before EC and how long did they use metformin before diagnosis by EC 

Author response: Thanks for the suggestion. 

Despite significant heterogeneity in this study, we performed a sensitivity analysis (excluding studies one by one) and determined that removing any one study did not affect the results. Our subgroup analysis indicates that the selection of reference materials, time bias, and confounding factors such as BMI and smoking may be sources of heterogeneity. We assessed the quality of evidence using GRADE, suggesting that our results possess a certain level of credibility. As a result of the limitations of the data in the included literatures, we were unable to determine the duration of diabetes before the diagnosis of esophageal cancer. We emailed the authors asking for details but received no response. We included the duration of metformin use in Table 1.

Reviewer #1: 1.Please indicate reference number for each study in table 1. Please provide the breakdown, indicating the number of patients/ participants on Metformin vs no Metformin. Currently on table 1, there is only one group as NM (Non-Metformin).

Author response: Thanks for the suggestion. 

We indicated the reference number for each study in table 1. We provided the breakdown, indicating the number of patients taking metformin and those not taking metformin.

2.What was the absolute number of new cases of esophageal cancer in the group exposed to Metformin and non-Metformin, in each study where incidence was the outcome of interest and combined.

Author response: We updated the absolute number of new esophageal cancer cases and incidence rate in the group exposed to Metformin and non-Metformin (Table 1).

3.What was the stage of the disease and histopathology in each study where a therapeutic intervention was done. 

Author response: We supplemented the disease stage and histopathological information (Table 1). The patients with esophageal cancer were primarily in the middle to advanced stages in the included studies. Our results indicate that metformin prolongs overall survival in patients with diabetes and esophageal cancer (especially esophageal squamous cell carcinoma). Details can be seen in Table 1.

4.Authors have provided significant statistical differences without any absolute numbers of an outcome/ event related to esophageal cancer or clinical characteristics of the patients exposed to Metformin or no Metformin, sulphonylureas or no sulphonylureas. 

Author response: We added absolute numbers of outcomes/events or clinical features related to esophageal cancer (Table 1: NO. of new cases of EC; Incidence rate; No. of deaths among patients with EC; Mortality rate).

5.Given the huge sample size in the analyzed studies and without detailed clinical information of the participants, incidence/ extent of esophageal cancer it is challenging to make a sense of statistical values provided in the manuscript.

Author response: We provided additional detailed clinical information for participants, including the number of new esophageal cancer cases, incidence rate, number of esophageal cancer deaths, mortality rate, disease staging, and tissue type (Table 1). We reached out to the authors of the included studies to obtain more patient information but received no response.

---

## [Editor Report · Decision Letter 1]

5 Sep 2024

Association of metformin use with risk and survival outcome of esophageal cancer in patients with diabetes: a systematic review and meta-analysis.

PONE-D-24-24173R1

Dear authors,

We’re pleased to inform you that your manuscript has been judged scientifically suitable for publication and will be formally accepted for publication once it meets all outstanding technical requirements.

Kind regards,

Yan Wang

Academic Editor

PLOS ONE

---

## [Editor Report · Acceptance letter]

7 Nov 2024

PONE-D-24-24173R1 

PLOS ONE

Dear Dr. Chen, 

I'm pleased to inform you that your manuscript has been deemed suitable for publication in PLOS ONE. Congratulations! Your manuscript is now being handed over to our production team.

Kind regards, 

on behalf of

Prof. Yan Wang 

Academic Editor

PLOS ONE